# In-Person and Online Studies Examining the Influence of Problem Solving on the Fading Affect Bias

**DOI:** 10.3390/bs14090806

**Published:** 2024-09-11

**Authors:** Jeffrey Alan Gibbons, Sevrin Vandevender, Krystal Langhorne, Emily Peterson, Aimee Buchanan

**Affiliations:** The College of Natural and Behavioral Sciences, Christopher Newport University, Newport News, VA 23606, USA; sevrin.vandevender.21@cnu.edu (S.V.); krystal.langhorne.21@cnu.edu (K.L.); emily.peterson.18@cnu.edu (E.P.); aimee.buchanan.18@cnu.edu (A.B.)

**Keywords:** fading affect bias, problem solving, in person, online

## Abstract

The fading affect bias (FAB) occurs in autobiographical memory when unpleasant emotions fade faster than pleasant emotions and the phenomenon appears to be a form of emotion regulation. As emotion regulation is positively related to problem solving, the current study examined FAB in the context of problem solving. In-person and online studies asked participants to provide basic demographics, describe their problem-solving abilities, and rate various healthy and unhealthy variables, including emotional intelligence and positive problem-solving attitudes. Participants also completed an autobiographical event memory form for which they recalled and described two pleasant and two unpleasant problem-solving and non-problem-solving events and rated the initial and current affect and rehearsals for those events. We found a robust FAB effect that was larger for problem-solving events than for non-problem-solving events in Study 1 but not in Study 2. We also found that FAB was positively related to healthy variables, such as grit, and negatively related to unhealthy variables, such as depression. Moreover, many of these negative relations were inverted at high levels of positive problem-solving attitudes, and these complex interactions were partially mediated by talking rehearsals and thinking rehearsals.

## 1. Introduction

From finding lost keys to fixing a car or a bad relationship or dealing with a prognosis of cancer, problems are an inevitable part of life, and humans either shrink or shine in their shadows. Left unresolved, problems fester and lead to psychological distress and negative affect [1]. Conversely, emotions influence problem solving, such that unpleasant emotions lead to poor problem-solving outcomes and pleasant emotions lead to successful problem-solving outcomes [2,3]. Based on the seemingly symbiotic relationship between emotions and problem solving, the fading affect bias (FAB), which is the faster fading of unpleasant than pleasant affect [4,5], should be a relevant mechanism that helps to predict and possibly account for the act of solving problems. The goal of the current study was to examine the relation of FAB to healthy and unhealthy variables in the context of problem solving.

### 1.1. Problem Solving and Emotions

The omnipresence of problems that people experience can be handled with optimism, and the creation and execution of plans for their resolution, which lead to positive perspectives about the experiences, or they can be addressed through pessimism and dread about the outcome of the problem and the difficulty of its solution, which lead to negative perspectives. Purposely or not, the action or inaction taken after a problem is the attempted solution and is intricately connected to emotional states. For example, the emotions of an 11-year-old male student learning geometry described by Näveri et al. (2011) [6] were analyzed as a case study by Hannula (2015) [7]. Specifically, the student incurred negative emotions, such as anger and shame, and performed poorly when experiencing peer ridicule, but he experienced positive emotions, such as excitement and joy, when he used humor to overcome negative emotional states.

Isen (2008) [8] argued that positive affect enhances the daily cognitive processes involved in problem solving, which was likely based on the finding in four studies that positive affect increased creative problem solving, whereas negative affect did not produce the same generative boost [9]. Positive affect seems to help people to overcome difficult challenges by facilitating flexibility [8]. Nelson and Sim (2014) [10] evaluated the effect of positive affect conditions compared to neutral (Experiment 1) and negative affect conditions (Experiment 2) on problem-solving ability in the context of hypothetical social problems. The researchers found that solutions received higher ratings in the positive affect conditions than the neutral affect condition in Experiment 1 (read or did not read statements designed to elicit a positive affective state) and the negative affect condition in Experiment 2 (read or listened to happy or sad media).

Instead of emphasizing the facilitating effects of pleasant affect in the context of problem solving, other researchers focused on the effects of negative affect. For example, Llera and Newman (2020 [11]) asked participants to provide a real-life problem that was affecting them that they could control, and then the researchers randomly assigned them to worry about the problem, think objectively when negative thoughts occurred, or perform diaphragmatic breathing. Participants were then asked to generate as many solutions as possible to their problem and then were rated on the effectiveness of their provided solutions. The results indicated that worry impaired participants’ problem-solving ability, indicating that negative emotions lead to poor problem solving.

Rather than using manipulations to create and evaluate group differences in impaired problem-solving ability, other researchers investigated poor problem-solving ability in established groups differing in negative emotions. For example, Marx and Schulze (1991) [12] used the Situation Specific Problem-Solving Inventory [13] to examine deficits in interpersonal problem-solving ability for depressed and non-depressed college students. The researchers found that depressed students displayed a greater deficit in their problem-solving ability compared to non-depressed students. Similarly, Korkmaz et al. (2020) [14] found diminished problem-solving abilities in 25 individuals who attempted suicide compared to the problem-solving abilities in 25 individuals who were not suicidal. However, suicide attempters with high emotional intelligence showed better problem-solving ability than suicide attempters with low emotional intelligence, which demonstrates the importance of skills related to expressing and interpreting emotions (i.e., emotional intelligence) for problem solving.

Instead of focusing on the effects of either positive or negative affect on problem-solving ability, some researchers were equally interested in both effects. For example, Fredrickson and Branigan (2005) [2] conducted two studies in which participants viewed films designed to evoke neutral, positive, and negative emotions. The researchers found that positive and negative emotions positively and negatively influenced cognition, respectively. Similarly, Lehman et al. (2008) [3] examined the emotions that occurred as participants attempted to solve logic problems and the role that affect plays in the problem-solving process. Participants were recorded as they solved logic problems and asked to retrospectively judge their affect by watching themselves back after they attempted to solve the presented problems. Curiosity, boredom, frustration, and happiness were the most prevalent emotions experienced by participants during the problem-solving process. Participants disengaged from problem solving when they experienced boredom but felt happy and were attentive when they successfully solved a problem. These results highlight the symbiotic relationship between problem solving and emotional affect.

### 1.2. The Fading Affect Bias (FAB)

The FAB is the faster fading of unpleasant than pleasant affect of autobiographical event memories [15]. The original research on this topic took place in the early 1900s, and it showed that participants recalled more pleasant events than unpleasant events [16,17,18,19], and that event affect faded quicker for unpleasant events than pleasant events [20]. Holmes (1970) [21] replicated both these findings and Walker et al. (1997) [4] replicated the differential fading of emotional event affect finding and they extended that finding across 3 months, 9 months, and 4.5 years, showing that the FAB increased with retention interval. Walker et al. [4] used Taylor’s (1991) [22] mobilization-minimization hypothesis to account for their findings. Specifically, biological, social, and cognitive resources are activated by and lessen the harmful effects of unpleasant events.

Gibbons and his colleagues found that the FAB occurred from 12 to 24 h, remained stable for up to 3 months [23], and did not differ across 8 to 12-year-old children and college students [24], even though the FAB was smaller for college students than 68 to 94-year-olds [25]. The FAB did not differ across alcohol and non-alcohol events [26], and the same non-difference was found across religious and non-religious events [27]. Whereas one study did not find FAB differences across events involving and not involving COVID-19 [28], another study did show that the FAB was smaller for events involving COVID-19 than events not involving COVID-19 [29]. In addition, the FAB was larger for events involving spirituality than for events not involving spirituality [27]. Compared to control events (e.g., non-social media events), the FAB was smaller for social media events [30], video game events [31], and political events including the presidential election [32]. These last results show that the FAB is smaller for emotionally unhealthy/hazardous events than for control events, which indicates that the FAB is a healthy outcome variable and possibly a healthy coping mechanism.

Additional research has provided data supporting the FAB as a healthy outcome variable because the FAB is positively related to healthy variables and negatively related to unhealthy variables. Healthy variables are positive, robust, and wholesome, whereas unhealthy variables are undesirable and unwanted (as suggested by Gibbons et al., 2023 [28]). The healthy variables include grit [28,33], social disclosures [34,35,36], social disclosures with a responsive listener [37,38], mature death attitudes [39], positive religious coping, and spirituality [27]. The FAB has also been positively related to self-esteem [30,40], partner esteem [40], and several relationship variables, such as relationship satisfaction and relationship confidence [40,41]. The unhealthy variables include dispositional mood [42], depression, anxiety, and stress [30,43,44], immature death attitudes [45], and engagement with social media [30]. Additional unhealthy variables include eating disorder symptoms [46], parental risk of physical abuse [47,48], and marijuana consumption [49].

Although the FAB was found to be larger for some religions, such as Buddhism and Judaism, than for other religions, such as Christianity [50], it did not differ across cultures [51]. Based on these results, Ritchie et al. (2014) [51] asserted that the FAB may be an evolutionary mechanism produced by a combination of biological, cognitive, and emotional resources that are activated and recruited to reduce the harmful effects of unpleasant event memories. The goal of this mobilization-minimization effect is to protect and enhance the integrity of an individual’s self by putting those unwelcomed events in perspective, which increases the seeking and avoidance of unpleasant and pleasant experiences, respectively [51,52].

The previously mentioned video game study showed that two unhealthy variables, including depression and Internet game addiction, combined to produce high FAB [31]. A gamer-generated explanation for the finding suggested that gamers are highly addicted to Internet games, and they experience frustration and depression when they first start a game, but also enjoy that novel experience due to the challenge, which only happens for new games. When the gamers figure out the games, they experience pride and boredom (i.e., low FAB) and move on to another game. Although these findings are somewhat strange, similar findings were demonstrated when comparing the FAB across COVID-19 anxiety as well as events involving and not involving COVID-19. Specifically, Gibbons et al. (2023) [28] found that several unhealthy variables, including hypochondria, neuroticism, time thinking and talking about COVID-19, current negative affect, and general anxiety, combined with COVID-19 anxiety to produce strong emotional regulation in the form of high FAB. The researchers explained that experiencing high levels of COVID-19 anxiety helped people to adapt to and come to expect and accept other unhealthy emotional states, such as general anxiety.

## 2. The Current Study

The problem-solving literature suggests that people tend to solve problems well when they experience pleasant emotional states. In addition, the video game study by Gibbons and Bouldin (2019) [31] suggested that the perplexing emotional states created by unsolved problems for expert problem solvers lead to high FAB. Similarly, unhealthy emotional states combined in the context of COVID-19 to produce high FAB, which may be the result of becoming an expert in dealing with COVID-19, just like someone addicted to Internet gaming. These similar findings from two different contexts suggest that problem solving should be strongly related to the FAB, but no study has examined the FAB in the context of problem solving. Therefore, the current study was created to fill this void in the literature and examine the relation of the FAB to healthy and unhealthy variables as well as problem-solving variables, such as problem-solving ability, and emotional intelligence across problem-solving and non-problem-solving events. The current study also tested the viability of rehearsal ratings as mediators of any significant, complex (three-way) interactions.

We expected the FAB to be higher for problem-solving events than for non-problem-solving events. We also expected the FAB to be negatively related to unhealthy variables, such as anxiety and depression, and we expected it to be positively related to healthy variables, such as problem-solving beliefs and emotional intelligence. Furthermore, we predicted that the relations between FAB and problem-solving beliefs and emotional intelligence would be stronger for problem-solving events than for non-problem-solving events, and we expected rehearsals to mediate these complex interactions. Finally, we predicted that the relations between FAB and continuous variables would increase with levels of positive problem-solving beliefs and emotional intelligence.

### 2.1. Study 1: In Person

Study 1 was conducted in person and participants filled out basic demographic questionnaires, a problem-solving ability questionnaire, and several other questionnaires including emotional intelligence and positive problem-solving attitudes. Participants also completed an autobiographical event memory form in which they recalled and described two pleasant and two unpleasant problem-solving and non-problem-solving events and rated the initial affect, current affect, and rehearsals for those events. Study 1 tested the hypotheses that were mentioned at the end of the general introduction.

#### 2.1.1. Method

##### Participants

The final sample of the current study consisted of responses from 284 participants who attended a small, public university in the southeastern United States. The participants for this study were recruited through Sona Systems (SONA), which is an online scheduling system used to record research participation credit and allow students to view and register for studies at their university. Data were collected from the fall of 2020 to the fall of 2022. After recruitment, participants were provided with a scheduled time for an in-person lab study. The participants included undergraduate students from a small, public, southeastern university who were enrolled in a psychology class. Participants received class credit for their participation. The demographic composition of the sample primarily included Caucasian (78%), Christian (63%) women (55%). The study received approval from the internal review board (IRB) of the university (IRB# 1558954-3, approved on 24 February 2020). Accordingly, participants were treated with the guidelines as specified by the American Psychological Association [53], which included briefing, consent, and debriefing. Participants were informed that they could leave the study at any time without penalty, and they were told that their data would be kept confidential under lock and key.

##### Materials and Measures

The materials included a consent form, which contained a briefing and a general description of the procedures, as well as contact information for the principal investigator, the IRB chair, and counseling services. The consent form also provided a place for participants to sign their names. The questionnaires assessed general demographic information (e.g., age, race, sex, religion, and sexual orientation) and the 40-item Mini Markers [54], which included our targeted neuroticism measure. The questionnaires also included the 20-item positive and negative affect schedule (PANAS; [55]), the brief, 21-item depression, anxiety, and stress scale (DASS; [56]), the 10-item Grit scale, the 19-item Pittsburgh Sleep Quality Index (PSQI; [57]), the 33-item Schutte Self Report Emotional Intelligence Test (SSEIT; [58]), the 30-item Inventory of Problem-Solving Abilities scale (IAPSA; [59]), which assessed attitudes about problem-solving abilities, and the self-reported, problem-solving and non-problem-solving events questionnaire. For each event, the event questionnaire asked for the date of event occurrence, a brief, four-line, event description, the initial event affect (at occurrence), and the final event affect (currently/at test), as well as three different rehearsal ratings.

The 40-item Mini Markers scale. The brief version of the Big Five personality factors was utilized as the first psychological measure in this study. This measure is also known as the 40-item Mini Markers Scale [54]. This item is designed to measure the participants’ openness, conscientiousness, extraversion, agreeableness, and neuroticism. For this study, however, only neuroticism was used. This sub-questionnaire lists various self-descriptive terms (e.g., energetic, imaginative, moody, bashful). Participants were asked to rate the extent to which they felt that these terms described themselves on a scale ranging from 1 (extremely inaccurate) to 9 (extremely accurate). Two items had to be reverse-scored, and then average neuroticism was calculated with high scores indicating high neuroticism. Cronbach’s alpha for neuroticism was 0.713.

Positive and Negative Affect Schedule (PANAS). The Positive and Negative Affect Schedule (PANAS; [55]) measures positive and negative affect with 20 questions about positive and negative emotions and the extent to which they have been felt by the participant in the last hour. Scores range from 1 (very slightly or not at all) to 6 (extremely). An example of one of the questions is “nervous” or “determined”. The items’ scores were averaged; Cronbach’s alpha for the Positive PANAS scale was 0.844, and Cronbach’s alpha for the Negative PANAS scale was 0.808.

Brief Depression, Anxiety, and Stress Survey (DASS-21). The brief Depression, Anxiety, and Stress Scale (DASS-21; [56]) was used to measure participants’ depression, anxiety, and stress, as these variables have been negatively related to the FAB in past research (e.g., [30]). The questionnaire included statements about depression, anxiety, and stress in which participants rated the strength that the statement applied to themselves. Scores ranged from 0 (did not apply to me at all) to 3 (applied to me very much or most of the time). An example statement is “I felt that life was meaningless”. Certain items pertain to stress, anxiety, or depression, and these certain items were added and scored with low scores indicating low levels of the relevant emotion. The scores from the items were averaged and Cronbach’s alpha for the depression portion of the DASS-21 scale was 0.882. Cronbach’s alpha for the anxiety portion of the DASS-21 scale was 0.762. Cronbach’s alpha for the stress portion of the DASS-21 scale was 0.738.

Grit. The Grit scale [60] uses 10 statements about grit. An example statement is “I have overcome setbacks to conquer an important challenge”. Participants indicated the degree that each statement resonated with them on a 5-point Likert-type scale with responses ranging from 1 (not at all) to 5 (very much). The odd-numbered items on the scale were asked in the opposite way as the even-numbered items. Therefore, the answers to the odd-numbered questions were reverse-scored and the average was calculated for the entire scale. Cronbach’s alpha for the GRIT scale was 0.811.

Sleep. The Pittsburgh Sleep Quality Index (PSQI; [57]) is a questionnaire with open-ended questions and Likert-type statements used to measure sleep quality and sleep disturbances. The open-ended questions approached the sleep duration with four questions, such as “during the past month, what time have you usually gone to bed at night?” Additionally, the seven closed-ended questions asked for the frequency of sleep disturbances with statements, such as, “during the past month, how often have you had trouble sleeping because you cannot get to sleep within 30 min?” Answers were given on a 4-point Likert-type scale ranging from 1 (Not during the past month) to 4 (Three or more times a week). Furthermore, participants were given a space to name any disturbance not named by the questionnaire, with a scale to provide the frequency that the item was experienced. The scores from the items were averaged and Cronbach’s alpha for the PSQI scale was 0.685.

Emotional intelligence. The Schutte Self-Report Emotional Intelligence Test [58] is a 33-item questionnaire that measures emotional intelligence. An example statement is “I have control over my emotions”. Responses ranged from scores of 1 (strongly disagree) to 5 (strongly agree). Certain items were reverse scored with low scores indicating high levels of emotional intelligence. The scores from the items were averaged and Cronbach’s alpha for the SSEIT was 0.815.

Positive problem-solving attitudes. The Inventory for Attitudes to Problem-Solving Ability (IAPSA; [59]) is a 30-item scale that measures attitudes to problem-solving ability. Participants indicate the extent to which they feel an item applies to them on a scale from 1 (never) to 5 (always). An example statement is “I can make an independent decision by myself”. Certain items were reverse scored with low scores indicating high attitudes to problem-solving ability. The scores from the items were averaged and Cronbach’s alpha for the IAPSA was 0.717.

Fading affect and rehearsal for events. The questionnaire prompted participants to describe eight events: two pleasant problem-solving events, two unpleasant problem-solving events, two pleasant non-problem-solving events, and two unpleasant non-problem-solving events. The order in which participants rated the different events was counterbalanced using a Latin square. As each kind of event, created by crossing event type and event affect, included two events, participants dated, described, and rated both events, and then they moved on to the next kind of event in the Latin square. Each event was rated for initial and current affect and rehearsals. Unpleasant events were initially rated on a single-item −3 (very unpleasant) to −1 (slightly unpleasant) scale and pleasant events were initially rated on a single-item 1 (slightly pleasant) to 3 (very pleasant) scale. The initial rating for pleasant events was positive, and the initial rating for unpleasant events was negative. Both unpleasant and pleasant events were rated for their current (at test) affect on the same single-item scale ranging from −3 (very unpleasant) to 3 (very pleasant). The current rating for an event could be positive, negative, or neutral. If event affect changed, participants wrote the amount of time that it took for that change to occur in days, hours, minutes, and seconds.

For initially pleasant events, fading affect was calculated by subtracting the current affect from the original affect. For initially unpleasant events, fading affect was calculated by subtracting initial affect from the current affect. These calculations ensured that positive fading affect meant that event affect reduced over time, whereas negative fading affect meant that event affect increased over time. Each event was also rated for the frequency that it was thought about, talked about, or both combined with a single-item scale ranging from 0 (never/infrequently) to 6 (always/very frequently). We initially examined fading affect for the 2267 events provided by participants but affect ratings and event descriptions were not provided for some events or event descriptions, or they were incorrectly rated, which resulted in 2121 remaining, usable events.

##### Procedure

Participants signed up for the study through SONA, and they were provided a timeslot to complete the in-person study. Upon arrival, participants received a consent form, in which they were informed that their consent was being sought for participation in the research study, their participation was completely voluntary, and they were allowed to stop the study at any time without any negative repercussions. Participants were also briefed on the fact that the study examined the emotions tied to pleasant and unpleasant event memories involving and not involving problem solving, as well as rehearsal ratings for these events. Participants were told that they should only provide events that do not cause emotional pain and, therefore, that the experimental procedure should provide no known risks to them.

Participants were informed that the experiment would demand up to 90 min of their time and that they would receive equitable class credit for their participation. Participants were told that the information that they provided was confidential and would only be examined by research assistants or the principal investigator of the study. Participants were also informed that their data were encrypted and placed under lock and key. Participants were then given the contact information of the principal investigator and the chair of the IRB. Participants were also given the contact information of the university counseling center to be used in the unlikely case that they experienced emotional discomfort. After receiving all the information in the briefing, participants signed the consent form before engaging in the procedure.

Following the briefing, participants completed a packet of questionnaires, which included general demographics, personality, mood, psychological distress, grit, emotional intelligence, positive problem-solving attitudes, and sleep quality. Then, participants recalled an event, provided information about the event, and then repeated that procedure for another event that was the same kind of event. The four kinds of event types included pleasant and unpleasant problem-solving and non-problem-solving events. Participants were told that problem-solving events involved an attempted solution (e.g., my car broke down, so I fixed it), and non-problem-solving events involved no attempted solution (e.g., simply drinking coffee). The participants were told that they had to describe events from their perspective and that the events had to involve the participant.

For each event, participants reported the date that the event occurred (as specific as possible), and they wrote a brief, four-line, event description disclosing as much information as they felt comfortable sharing. Participants then provided an initial/original emotion for the way that they felt at the time of the event. Participants were told that unpleasant events should initially be rated using a negative number ranging from −3 (very unpleasant) to −1 (mildly unpleasant) and pleasant events should be initially rated using a positive number ranging from 1 (mildly pleasant) to 3 (very pleasant). Participants were asked to rate the way that they currently (at test) felt about the event, and they rated their current emotion on a scale ranging from −3 (very unpleasant) to +3 (very pleasant). Participants then provided the frequency that they thought about the event, talked about it, and both thought and talked about it on a rating scale ranging from 0 (never/infrequently) to 6 (always/very frequently). Finally, participants rated whether event affect had stopped changing, and they recorded the amount of time in days, hours, minutes, and seconds required for that change to finish, or they could mark that their event affect was still changing. Finally, participants were given a debriefing form, were asked to read it in its entirety, and were asked if they had any questions. Participants were given credit through SONA following their debriefing.

The four kinds of events were presented in an order that was counterbalanced using a Latin square, which created four order conditions. The first order consisted of two pleasant problem-solving events, two pleasant non-problem-solving events, two unpleasant problem-solving events, and two unpleasant non-problem-solving events. The second order consisted of two pleasant non-problem-solving events, two unpleasant non-problem-solving events, two pleasant problem-solving events, and two unpleasant problem-solving events. The third order was the reverse of the second order and the fourth order was the reverse of the first order. Similarly, the rehearsal ratings were presented in one of two orders. Specifically, the combined thinking and talking rehearsals occurred either first or last with the individual rehearsals always beginning with thinking rehearsals followed by talking rehearsals. These counterbalancing controls created eight different order conditions in the study.

##### Analytic Strategy

For each analysis, event was the unit of analysis. Unusable data were removed and not analyzed. We first tested the two-way interaction of initial event affect (pleasant vs. unpleasant) and event type (problem-solving and non-problem solving) on initial affect intensity and fading affect using a 2 (initial event affect) × 2 (event type) completely between-groups design with initial event affect (pleasant or unpleasant) and event type (problem-solving or non-problem-solving) as the independent variables. However, we used an analysis of variance (ANOVA) to statistically evaluate initial affect intensity, whereas we used an analysis of covariance (ANCOVA) with initial affect intensity as the covariate to statistically evaluate fading affect. Follow-up independent group t-tests were used to examine interactions if they were significant. We then employed the Process macro via IBM SPSS [61] to test for two-way and three-way interactions involving initial event affect and continuous variables as predictors of fading affect while controlling for initial affect intensity. For any statistically significant finding for these interaction analyses produced by the Process macro, we reported the indirect effect, the corresponding standard error, t-value, *p*-value, and 95% CI lower- and upper-estimates, as well as effect size at each level of the moderators.

We used Model 1 of the Process macro to examine fading affect, y, across initial event affect, x, conditional upon levels of self-reported individual difference variables, w. These variables included neuroticism, positive and negative PANAS, depression, anxiety, stress, grit, poor sleep via the PSQI, talking rehearsals, thinking rehearsals, and talking and thinking rehearsals, as well as self-reported emotional intelligence (SSEIT), and problem-solving ability attitudes (IAPSA). In Process Model 1, we controlled a nominal-level participant variable to control for clustered data and initial affect intensity to control for regression to the mean in each model. We used the Johnson–Neyman technique to indicate where along an individual difference variable the FAB was weak or strong, which avoids drawing an arbitrary line to determine “low” and “high” groups (Preacher et al., 2006 [62]).

To test for any significant three-way interactions, we again utilized the Process macro to examine fading affect, y, among four categories of events across the spectrum of the individual difference variables. Specifically, Model 3 enabled the specification of the two-way interaction between initial event affect, x, and event type or positive problem-solving attitudes (IAPSA) or emotional intelligence (SSEIT), m, while controlling for the nominal-level participant variable and initial affect intensity, conditional upon levels of self-reported individual difference variables, w. These variables included neuroticism, positive and negative PANAS, depression, anxiety, stress, grit, and poor sleep via the PSQI. We examined fading affect across the spectrum of individual difference variables for each of the four events (pleasant and unpleasant problem-solving and non-problem-solving) using the Johnson–Neyman technique. We also examined the interactive effect of initial event affect and positive problem-solving attitudes (IAPSA) or emotional intelligence (SSEIT) using the Johnson–Neyman technique. The goal of these analyses was to indicate the exact value for the variable where (1) the effect of event type on FAB was large and small, and (2) the relation of positive problem-solving attitudes (IAPSA) or emotional intelligence (SSEIT) was strong and weak.

We also evaluated possible mediators of the three-way interaction with the Process macro. We examined each of the three event rehearsal frequency ratings as a mediator of significant relations between fading affect and initial event affect (i.e., FAB) to individual difference variables across event type or positive problem-solving attitudes (IAPSA) or emotional intelligence (SSEIT). Process Model 11 enables the replication of the three-way interaction (i.e., Model 3 is tested within Model 11), as well as evaluations of the mediators for this effect. In Model 11, we hypothesized that initial event affect (unpleasant vs. pleasant), x, would interact with event type (problem-solving and non-problem solving), z, or positive problem-solving attitudes or emotional intelligence, z, to predict fading affect, y, across levels of individual difference variables, w, and this effect of x*w*z on y may occur through event rehearsal frequency, m. We reported the conditional indirect effect of x*w*z on y through m, examining the indirect effect of x on y through m at levels of the moderators, w and z. In each model, we tested for mediation in any significant three-way interactions, controlling for the potential influence of participants.

#### 2.1.2. Results

##### Discrete Two-Way Interactions

The ANOVA for initial affect intensity produced heterogeneity, but this parametric assumption violation is not a problem if the sample sizes are relatively equal, defined by a ratio of largest to smallest sample sizes equal to or less than 1.5 [63]. The sample size ratios calculated for initial event affect, event type, and the ratios were all less than 1.5, and therefore relatively equal. The overall analysis of variance investigating initial affect intensity was statistically significant, *F*(3, 2117) = 49.387, *p* < 0.001, η_p_^2^ = 0.065 (Figure 1). Pleasant events (*M* = 2.330, *SE* = 0.023) were initially more intense than unpleasant events (*M* = 2.418, *SE* = 0.023), *F*(1, 2117) = 7.475, *p* < 0.01, η_p_^2^ = 0.004, which does not support regression to the mean as an explanation for FAB effects. The problem-solving events (*M* = 2.242, *SE* = 0.024) were initially less intense than the non-problem events (*M* = 2.506, *SE* = 0.022), *F*(1, 2117) = 66.859, *p* < 0.001, η_p_^2^ = 0.031.

The initial event affect × event type interaction was statistically significant for initial affect intensity, *F*(1, 2117) = 74.397, *p* < 0.001, η_p_^2^ = 0.034, and additional analyses were conducted to break down this interaction. For problem-solving events, initial affect intensity was larger for unpleasant (*M* = 2.425, *SE* = 0.032) than pleasant events (*M* = 2.059, *SE* = 0.035), *t*(1055) = −7.697, *p* < 0.001, η_p_^2^ = 0.053, but initial affect intensity was smaller for unpleasant (*M* = 2.411, *SE* = 0.032) than pleasant (*M* = 2.600, *SE* = 0.029) non-problem-solving events, *t*(1062) = 4.362, *p* < 0.001, η_p_^2^ = 0.018. These results indicate that regression to the mean could potentially explain FAB effects for problem-solving events. Therefore, we statistically controlled for initial affect intensity when conducting all the analyses with fading affect as the dependent variable.

When analyzing FAB effects with fading affect as the dependent variable and both participant and initial affect intensity as the covariates using ANCOVA, heterogeneity was found but it was not an issue for the same reasons mentioned previously for the ANOVA examining initial affect intensity. The overall ANOVA was statistically significant, *F*(3, 2116) = 363.994, *p* < 0.001, η_p_^2^ = 0.408 (Figure 2). Pleasant events (*M* = 0.050, *SE* = 0.034) showed lower fading affect than unpleasant events (*M* = 2.001, *SE* = 0.055), *F*(1, 2116) = 1048.833, *p* < 0.001, η_p_^2^ = 0.331, which demonstrated a robust FAB. Non-problem solving events (*M* = 0.872, *SE* = 0.043) showed lower fading affect than problem-solving events (*M* = 1.179, *SE* = 0.043), *F*(1, 2116) = 25.220, *p* < 0.001, η_p_^2^ = 0.012. The initial event affect × event type was statistically significant, *F*(1, 2116) = 8.740, *p* < 0.01, η_p_^2^ = 0.004. Additional analyses breaking down this interaction showed that the fading affect bias (greater fading affect for unpleasant than pleasant events) was larger for problem-solving events, *t*(1054) = −22.560, *p* < 0.001, η_p_^2^ = 0.326, than non-problem-solving events, *t*(1061) = −22.231, *p* < 0.001, η_p_^2^ = 0.318. However, the two events showed small differences in the magnitude of the *t*-values and the FAB effect sizes.

##### Continuous Two-Way Interactions

As previously stated, we statistically controlled for initial affect intensity when conducting the analyses, with fading affect as the dependent variable. We used Process Model 1 [61] to examine whether individual difference variables predicted the FAB. These variables included positive PANAS, emotional intelligence, positive problem-solving attitudes, talking rehearsal ratings, thinking rehearsal ratings, thinking and talking rehearsal ratings, grit, neuroticism, negative PANAS, depression, anxiety, stress, and poor sleep. The continuous variables that predicted FAB included talking rehearsal ratings, thinking and talking rehearsal ratings, grit, positive PANAS, emotional intelligence, positive problem-solving attitudes, depression, anxiety, and stress. When observing positive PANAS as a predictor for the FAB, all main effects (initial event affect, positive PANAS, initial affect intensity, and participant) were significant. In addition, the results from Process Model 1 [61] revealed a significant two-way interaction between positive PANAS and initial event affect; B = 0.511 (*SE* = 0.076), *t*(2216) = 6.734, *p* < 0.001, 95% CI [0.362, 0.660], Model Δ*R*^2^ (due to the two-way interaction) = 0.012, overall Model *R*^2^ = 0.409, *p* < 0.001. Figure 3 displays that the FAB increased with positive PANAS because fading affect increased for unpleasant events as positive PANAS increased.

When evaluating emotional intelligence (SSEIT) as a predictor for the FAB, the main effects of initial event affect, SSEIT, initial affect intensity, and participant were significant. Additionally, the results from Process Model 1 [61] revealed a significant two-way interaction between SSEIT and initial event affect; B = 0.883 (*SE* = 0.156), *t*(2197) = 5.672, *p* < 0.001, 95% CI [0.577, 1.188], Model Δ*R*^2^ (due to the two-way interaction) < 0.009, overall Model *R*^2^ = 0.402, *p* < 0.001. The FAB increased with SSEIT because fading affect decreased for pleasant events as SSEIT increased (i.e., Figure 3). When observing positive problem-solving attitudes (IAPSA) as a predictor for the FAB, the main effects of IAPSA, initial affect intensity, and participant were significant. In addition, the results from Process Model 1 [61] revealed a significant two-way interaction between IAPSA and initial event affect; B = 0.776 (*SE* = 0.202), *t*(2094) = 3.840, *p* < 0.001, 95% CI [0.380, 1.172], Model Δ*R*^2^ (due to the two-way interaction) = 0.004, overall Model *R*^2^ < 0.402, *p* < 0.001. The FAB increased with IAPSA, because fading affect decreased for pleasant events as IAPSA increased (i.e., Figure 3).

When examining talking rehearsals as a predictor of the FAB, the main effects of talking rehearsals, initial event affect, and initial affect intensity were each significant. More importantly, the results from Process Model 1 [61] revealed a significant two-way interaction between talking rehearsals and initial event affect; B = 0.103 (*SE* = 0.032), *t*(2218) = 3.185, *p* < 0.002, 95% CI [0.040, 0.166], Model Δ*R*^2^ (due to the two-way interaction) < 0.003, overall Model *R*^2^ = 0.396, *p* < 0.001. The FAB increased with talking rehearsals because fading affect decreased for pleasant events and increased for unpleasant events as talking rehearsals increased (i.e., Figure 3). When examining both thinking and talking rehearsals combined as a predictor of the FAB, the main effects of the combined rehearsals, initial event affect, initial affect intensity, and participants were each significant. More importantly, the results from Process Model 1 [62] revealed a significant two-way interaction between the combined rehearsals and initial event affect; B = 0.075 (*SE* = 0.034), *t*(2217) = 2.226, *p* < 0.027, 95% CI [0.009, 0.141], Model Δ*R*^2^ (due to the two-way interaction) < 0.002, overall Model *R*^2^ = 0.396, *p* < 0.001. The FAB increased with the combined rehearsals because fading affect decreased for pleasant events and increased for unpleasant events as the combined rehearsals increased (i.e., Figure 3).

When analyzing grit as a predictor of the FAB, the main effects for initial event affect, grit, and initial affect intensity were significant. In addition, the results from Process Model 1 (Hayes, 2022 [61]) revealed a significant two-way interaction between grit and initial event affect; B = 0.230 (*SE* = 0.094), *t*(2216) = 2.442, *p* < 0.015, 95% CI [0.045, 0.416], Model Δ*R*^2^ (due to the two-way interaction) < 0.002, overall Model *R*^2^ = 0.395, *p* < 0.001. The FAB increased with grit because fading affect decreased for pleasant events as grit increased (i.e., Figure 3).

When examining depression as a predictor of the FAB, the main effect of depression and initial event affect were significant. Moreover, the results from Process Model 1 (Hayes, 2022 [61]) revealed a significant two-way interaction between depression and initial event affect; B = −0.591 (*SE* = 0.089), *t*(2211) = −6.634, *p* < 0.001, 95% CI [−0.765, −0.416], Model Δ*R*^2^ (due to the two-way interaction) < 0.012, overall Model *R*^2^ < 0.406, *p* < 0.001. Figure 4 shows that the FAB decreased with depression because fading affect decreased for pleasant events, and it increased for unpleasant events as depression increased. When examining anxiety as a predictor of the FAB, the main effects of initial event affect, anxiety, and initial affect intensity were significant. Importantly, the results from Process Model 1 [61] revealed a significant two-way interaction between anxiety and initial event affect; B = −0.323 (*SE* = 0.094), *t*(2219) = −3.428, *p* < 0.001, 95% CI [−0.508, −0.138], Model Δ*R*^2^ (due to the two-way interaction) > 0.003, overall Model *R*^2^ = 0.396, *p* < 0.001. The FAB increased with anxiety, primarily because fading affect decreased for unpleasant events as anxiety increased (i.e., Figure 4).

When examining stress as a predictor of the FAB, the main effects of initial event affect, stress, and initial affect intensity were significant, but the main effect of participants was not significant. Moreover, the results from Process Model 1 [61] revealed a significant two-way interaction between stress and initial event affect; B = −0.394 (*SE* = 0.102), *t*(2216) = −3.859, *p* < 0.001, 95% CI [−0.594, −0.194], Model Δ*R*^2^ (due to the two-way interaction) = 0.004, overall Model *R*^2^ = 0.398, *p* < 0.001. The FAB decreased with stress primarily because fading affect decreased for unpleasant events as stress increased (i.e., Figure 4).

##### Continuous Three-Way Interactions

To test for significant three-way interactions, we used the Process macro to examine fading affect, *y*, among four categories of events across the continuum of the problem-solving and other individual difference variables. Specifically, Model 3 [61] enabled the specification of the two-way interaction between initial event affect, *x*, and individual difference variables, *m*, while controlling for participant and initial affect intensity, conditional upon event type, levels of self-reported positive problem-solving attitudes (IAPSA), or emotional intelligence (SSEIT), *w*. We also used the Johnson–Neyman technique to detect where the FAB was more strongly related to an individual difference variable (i.e., positive PANAS) for one event type (e.g., problem-solving event) than for another event type, or a particular point on the positive problem-solving attitudes (IAPSA) scale or the emotional intelligence (SSEIT) scale.

We used the Hayes Process Model 3 (Hayes, 2022 [61]) to examine the three-way interaction of initial event affect, positive PANAS, and emotional intelligence (SSEIT) while controlling for initial affect intensity and participant. The model revealed significant main effects of initial event affect, initial affect intensity, and participant, as well as a significant two-way interaction between positive PANAS and initial event affect. Moreover, a significant three-way interaction was found between emotional intelligence (SSEIT), positive PANAS, and initial event affect; B = 0.551 (*SE* = 0.195), *t*(2189) = 2.831, *p* < 0.005, 95% CI [0.169, 0.932], Model Δ*R*^2^ (due to the three-way interaction) > 0.002, overall Model *R*^2^ = 0.421, *p* < 0.001. Figure 5a–e and the Johnson–Neyman values showed a non-significant negative relation between FAB and positive PANAS at and after the first quintile of emotional intelligence (SSEIT). Afterward, the relation inverted forming a positive relation between FAB and positive PANAS that became significant before the second quintile of SSEIT at a rating of 3.253 and increased with SSEIT from that point.

When examining the three-way interaction between initial event affect, neuroticism, and positive problem-solving beliefs (IAPSA), all the main effects were significant in the Hayes Process Model 3 (Hayes, 2022 [61]) with the exception of IAPSA. In addition, the two-way interactions were significant with the exception of the initial event affect by the IAPSA interaction. More importantly, Model 3 revealed a significant three-way interaction between positive problem-solving attitudes (IAPSA), initial event affect, and neuroticism; B = 0.384 (*SE* = 0.159), *t*(2058) = 2.420, *p* < 0.02, 95% CI [0.073, 0.696], Model Δ*R*^2^ (due to the three-way interaction) < 0.002, overall Model *R*^2^ = 0.404, *p* < 0.05. Figure 6a–e and the Johnson–Neyman values initially demonstrated a significant negative relation between FAB and neuroticism that decreased as IAPSA increased and was last significant at IAPSA ratings of 3.434, and that negative relation inverted at IAPSA ratings of 3.792 and increased, but did not become significant, from that point.

##### Examining Rehearsals as Mediators of the Three-Way Interactions

Next, we found and examined the conditional indirect effects of initial event affect on fading affect for (1) positive PANAS across emotional intelligence (SSEIT) and (2) neuroticism across levels of positive problem-solving attitudes (IAPSA) through rehearsal ratings (talking, thinking, thinking and talking) using the Process Model 11 [61]. The three-way interaction involving fading affect, initial event affect, positive PANAS, and emotional intelligence (SSEIT) was intervened by talking rehearsals at the second and third quintiles of positive PANAS across the second and third quintiles of SSEIT. This same interaction was intervened by thinking rehearsals at the first quintiles of both positive PANAS and SSEIT, and the first and second quintiles of positive PANAS across the second and third quintiles of SSEIT. This interaction was also intervened by thinking and talking rehearsals at the fourth and fifth quintiles of positive PANAS at the first quintile of SSEIT, the second, third, and fourth quintiles of positive PANAS across the second and third quintiles of SSEIT, and the third quintile of positive PANAS at the fourth quintile of SSEIT.

The interaction involving fading affect, initial event affect, neuroticism, and positive problem-solving beliefs (IAPSA) was intervened by talking rehearsals at the second and third quintiles of neuroticism across the third quintile of IAPSA. This same interaction was intervened by thinking rehearsals at the fourth and fifth quintiles of neuroticism across the first and second quintiles of IAPSA, as well as the fifth quintile of neuroticism at the third quintile of IAPSA. The interaction was also intervened by thinking and talking rehearsals at the first and second quintiles of neuroticism across the first, second, and third quintiles of IAPSA.

#### 2.1.3. Discussion

Study 1 demonstrated a significant fading affect bias (FAB) effect, which is the faster fading of unpleasant than pleasant affect, and it was larger for problem-solving events than for non-problem-solving events. These results replicate past FAB research (e.g., [4,28,51]) and they extend it to the context of problem-solving. In addition, the FAB was positively predicted by both talking rehearsals and the combination of talking and thinking rehearsals with a slightly stronger effect for talking rehearsals alone. This finding supported previous FAB research showing that verbal event sharing is an effective mood regulation method [34,35,36]. Other healthy variables, including positive PANAS, emotional intelligence, positive problem-solving attitudes, and grit positively predicted FAB, and many unhealthy measures, such as depression, anxiety, and stress, negatively predicted FAB. These findings support and extend previous research (e.g., [28.49]). These findings also support assertions that the emotional damage caused by unpleasant autobiographical events activates biological, cognitive, and emotional resources that counteract it [22] and motivate people to seek out pleasant experiences and avoid unpleasant ones, which enhances self-perceptions [51,52].

Two significant three-way interactions were found. The first three-way interaction showed that positive PANAS and emotional intelligence combined to predict the FAB. More specifically, positive PANAS positively predicted the FAB, but this relation only became significant near the second quintile of emotional intelligence, and it increased from that point. This result displays that emotion regulation (i.e., FAB) produced by good feelings at the moment only manifested for individuals at or above medium-low levels of emotional intelligence. This finding highlights the importance of emotional intelligence for emotion regulation. Furthermore, the effects of positive problem-solving attitudes were so powerful that they completely changed the negative relation of neuroticism and FAB present at low levels of positive problem-solving attitudes into a positive relation at high levels of positive problem-solving attitudes. Positive problem-solving attitudes curbed the poor emotion regulation that typically occurs with high levels of neuroticism and turned it into highly positive and healthy emotion regulation.

All three rehearsal rating types mediated the two three-way interactions. However, talking rehearsals mediated the three-way interactions at 6 of the 50 possible quintiles, thinking rehearsals mediated them at 10 of the 50 quintiles, and both thinking and talking rehearsals mediated them at 15 of the 50 quintiles. These results show that thinking rehearsals unexpectedly explained complex effects more effectively than talking rehearsals, but less effectively than the combined thinking and talking rehearsal measure rating, which was expected based on past research [26,27,31]

### 2.2. Study 2: Online Replication

Study 2 replicated Study 1, except it was conducted online rather than in person, and it was conducted during the COVID-19 pandemic. As Study 1 was conducted in person preceding and following the pandemic, it became a control for Study 2. More importantly, the procedural differences in the two studies allowed for strong tests of the internal and external validity of the results in Study 1. Specifically, similar results across the two studies indicate both internal validity in the form of statistically reliable results, and external validity as the results generalize across different procedures (in person vs. online) and times (before and after the pandemic vs. during the pandemic). Conversely, different results do not demonstrate validity. Like Study 1, we expected a robust FAB effect, with a larger FAB for problem-solving events than non-problem-solving events. Based on the results of Study 1, we expected FAB to be negatively predicted by unhealthy variables and positively predicted by healthy variables. We also expected three-way interactions involving emotional intelligence and positive problem-solving attitudes.

#### 2.2.1. Method

##### Participants

The final sample included 157 students from a small, public university in the southeastern part of the United States. Individuals were gathered online using Qualtrics, which was made accessible for participants through SONA. The participants were all 18 years of age or older. The participants were primarily Caucasian (76%), Christian (67%), and women (67%). The study received IRB approval (IRB# 1610883-1, approved on 23 October 2020), and they followed APA (2023) [53] ethical guidelines, which included a briefing, signed consent, and a debriefing.

##### Materials and Measures

The materials used in Study 2 were the same ones used in Study 1. Similarly, the measures derived from the materials in Study 2 were the same as the ones used in Study 1.

Cronbach’s alpha for neuroticism was 0.736. The Cronbach’s alphas for positive PANAS and negative PANAS were 0.896 and 0.842, respectively. Cronbach’s alphas for depression, anxiety, and stress were 0.792, 0.883, and 0.767, respectively. The Cronbach’s alphas for grit and PSQI (poor sleep) were 0.750 and 0.758, respectively. Cronbach’s alpha for the IAPSA was 0.907.

##### Procedure and Analytic Strategy

The method used for Study 2 mirrored the method from Study 1, except that Study 1 was conducted in person before and after the COVID-19 pandemic, whereas Study 2 was conducted online using Qualtrics during the pandemic. Both studies tested college-aged participants from the same university. The analytic strategy used in Study 2 was the same as the one used in Study 1. Event was the unit of analysis in Study 2 as it was in Study 1. Study 2 included 1087 events in its analyses.

#### 2.2.2. Results

##### Discrete Two-Way Interactions

The ANOVA for initial affect intensity and the ANCOVA for fading affect produced heterogeneity, but this violation of this parametric assumption for conducting ANOVA and ANCOVA is not a problem if the sample sizes are relatively equal, defined by a ratio of largest to smallest sample sizes equal to or less than 1.5 [63]. The sample size ratios calculated for initial event affect, event type, and their interaction for both measures were all less than 1.5, and, therefore, relatively equal. When investigating initial affect intensity, the overall ANOVA was statistically significant, *F*(3, 1046) = 22.906, *p* < 0.001, η_p_^2^ = 0.062 (Figure 7). Pleasant events (*M* = 2.314, *SE* = 0.036) demonstrated lower initial affect intensity than unpleasant events (*M* = 2.422, *SE* = 0.032), *F*(1, 1046) = 5.249, *p* = 0.022, η_p_^2^ = 0.005, which indicated that regression-to-the mean could potentially explain any FAB effects. Non-problem-solving events (*M* = 2.486, *SE* = 0.033) displayed lower initial affect intensity than problem-solving events (*M* = 2.257, *SE* = 0.035), *F*(1, 1046) = 26.104, *p* < 0.001, η_p_^2^ = 0.024. The initial event affect × event type was statistically significant, *F*(1, 1046) = 39.013, *p* < 0.001, η_p_^2^ = 0.036. The data pattern describing the two-way interaction for initial affect intensity in Study 2 mirrored the pattern in Study 1, and it indicated that regression-to-the-mean could explain any FAB effects for problem-solving events. Therefore, we controlled for initial affect intensity when examining FAB effects.

As initial affect intensity was higher for unpleasant than pleasant events indicating that regression-to-the-mean could account for the FAB effects, we ran the initial 2 (initial event affect) × 2 (event type) ANCOVA with fading affect as the dependent variable and initial affect intensity as the covariate. The overall analysis of covariance (ANCOVA) investigating fading affect was statistically significant, *F*(4, 1045) = 123.156, *p* < 0.001, η_p_^2^ = 0.320. Pleasant events (*M* = 0.416, *SE* = 0.060) showed lower fading affect than unpleasant events (*M* = 2.243, *SE* = 0.079), *F*(1, 1045) = 336.995, *p* < 0.001, η_p_^2^ = 0.244, which demonstrated strong fading affect bias with initial affect intensity controlled. Non-problem-solving events (*M* = 1.304, *SE* = 0.073) showed lower fading affect than problem-solving events (*M* = 1.405, *SE* = 0.088), *F*(1, 1045) = 5.326, *p* = 0.021, η_p_^2^ = 0.005, when initial affect intensity was controlled. In addition, the initial event affect × event type interaction was not statistically significant, *F*(1, 1045) = 0.784, *p* > 0.05, η_p_^2^ = 0.001, when initial affect intensity was statistically controlled (Figure 8).

##### Continuous Two-Way Interactions

We used Process Model 1 [61] to examine whether individual difference variables predicted fading affect while controlling for the nominal-level participant variable and initial affect intensity. These variables included talking rehearsal ratings, thinking rehearsal ratings, thinking and talking rehearsal ratings, grit, emotional intelligence, positive problem-solving attitudes, positive PANAS, negative PANAS, depression, anxiety, stress, and poor sleep. When examining talking rehearsals as a predictor of the FAB, all main effects (initial event affect, talking rehearsals, initial affect intensity, and participant) were significant. More importantly, the results from Process Model 1 [61] revealed a significant two-way interaction between talking rehearsals and initial event affect; B = 0.236 (*SE* = 0.049), *t*(1031) = 4.779, *p* < 0.001, 95% CI [0.139, 0.333], Model Δ*R*^2^ (due to the two-way interaction) < 0.015, overall Model *R*^2^ = 0.340, *p* < 0.001. Figure 9 shows that the FAB increased with talking rehearsals because fading affect decreased for pleasant events and increased for unpleasant events as talking rehearsals increased.

When examining thinking and talking rehearsals as a predictor of the FAB, all main effects (initial event affect, thinking and talking rehearsal, initial affect intensity, and participant) were significant. More importantly, the results from Process Model 1 [61] revealed a significant two-way interaction between thinking and talking rehearsals and initial event affect; B = 0.227 (*SE* = 0.049), *t*(1031) = 4.643, *p* < 0.001, 95% CI [0.131, 0.323], Model Δ*R*^2^ (due to the two-way interaction) < 0.014, overall Model *R*^2^ = 0.340, *p* < 0.001. The results demonstrated that the FAB increased with thinking and talking rehearsals mainly because fading affect decreased for pleasant events (i.e., Figure 9). When examining emotional intelligence (SSEIT) as a predictor of the FAB, all the main effects, except initial event affect, were significant. Moreover, the results from Process Model 1 [61] revealed a significant two-way interaction between emotional intelligence (SSEIT) and initial event affect; B = 0.495 (*SE* = 0.209), *t*(1034) = 2.367, *p* = 0.018, 95% CI [0.085, 0.905], Model Δ*R*^2^ (due to the two-way interaction) < 0.004, overall Model *R*^2^ = 0.325, *p* < 0.001. The results show that the FAB increased with emotional intelligence (SSEIT) because fading affect decreased for pleasant events (i.e., Figure 9).

When examining neuroticism as a predictor of the FAB, all the main effects, except neuroticism, were significant. More importantly, the results from Process Model 1 [61] revealed a significant two-way interaction between neuroticism and initial event affect; B = 0.190 (*SE* = 0.088), *t*(1020) = 2.146, *p* = 0.032, 95% CI [0.016, 0.363], Model Δ*R*^2^ (due to the two-way interaction) = 0.003, overall Model *R*^2^ = 0.329, *p* < 0.001. Figure 10 displays that the FAB increased with neuroticism primarily because fading affect increased for unpleasant events as neuroticism increased. When examining poor sleep as a predictor of FAB, all the main effects were significant, except poor sleep. Moreover, the results from Process Model 1 [61] revealed a two-way interaction between poor sleep and initial event affect that approached significance; B = −0.432 (*SE* = 0.221), *t*(1024) = −1.956, *p* < 0.051, 95% CI [−0.865, 0.001], Model Δ*R*^2^ (due to the two-way interaction) < 0.003, overall Model *R*^2^ = 0.320, *p* < 0.001. The results show that the FAB decreased with poor sleep because fading affect decreased for unpleasant events and increased for pleasant events as the combined rehearsals increased (i.e., Figure 10).

##### Continuous Three-Way Interactions

To test for significant three-way interactions, we again used the Process macro to examine fading affect, y, among four categories of events across the continuum of the problem-solving variables and other individual difference variables. Specifically, Model 3 [61] enabled the specification of the two-way interaction between initial event affect, x, and individual difference variables, m, while controlling for participant and initial affect intensity, conditional upon event type, levels of self-reported positive problem-solving attitudes (IAPSA), or emotional intelligence (SSEIT), w. We also used the Johnson–Neyman technique to detect where the FAB was more strongly related to an individual difference variable (i.e., neuroticism) for one event type (e.g., problem-solving event) than for another event type (e.g., non-problem-solving event), or a particular point on the positive problem-solving attitudes (IAPSA) scale or the emotional intelligence (SSEIT) scale.

When examining the three-way interaction of initial event affect, positive problem-solving attitudes (IAPSA), and neuroticism on fading affect while controlling for initial affect intensity and participant, the Hayes Process Model 3 (Hayes, 2022 [61]) revealed that all the main effects and two-way interactions were significant. Moreover, the model revealed a significant three-way interaction between positive problem-solving attitudes (IAPSA), neuroticism, and initial event affect; B = 0.515 (*SE* = 0.231), *t*(833) = 2.227, *p* < 0.027, 95% CI [0.061, 0.968], Model Δ*R*^2^ (due to the three-way interaction) < 0.004, overall Model *R*^2^ = 0.352, *p* < 0.001. The Johnson–Neyman values showed that no relation between FAB and neuroticism was found for the first two quintiles of positive problem-solving attitudes (IAPSA), but the positive relation between FAB and neuroticism became significant after the third quintile of IAPSA and increased with IAPSA from that point (i.e., Figure 11a–e). This result replicates the pattern from Study 1 for this same three-way interaction.

When examining the three-way interaction of initial event affect, positive problem-solving attitudes (IAPSA), and depression on fading affect while controlling for initial affect intensity and participant, the Hayes Process Model 3 [61] revealed that all the main effects and two-way interactions were significant. The Hayes model revealed a significant three-way interaction between positive problem-solving attitudes (IAPSA), depression, and initial event affect; B = 1.692 (*SE* = 0.421), *t*(841) = 4.018, *p* < 0.001, 95% CI [0.865, 2.518], Model Δ*R*^2^ (due to the three-way interaction) = 0.012, overall Model *R*^2^ = 0.360, *p* < 0.001. Figure 11a–e and the Johnson–Neyman values demonstrated a negative relation between FAB and depression that began significant and decreased as IAPSA increased until it stopped being significant at IAPSA ratings of 3.083 (just below the second quintile), and then the relation flipped and became significantly positive at IAPSA ratings of 3.516 (just below the fourth quintile).

The Hayes Process Model 3 (Hayes, 2022 [61]) examined the three-way interaction of initial event affect, IAPSA, and anxiety on fading affect while controlling for initial affect intensity and participant. The model revealed that all the main effects and two-way interactions were significant, except for the main effects of anxiety and IAPSA and the two-way interaction between anxiety and IAPSA. The model revealed a significant three-way interaction between positive problem-solving attitudes (IAPSA), anxiety, and initial event affect; B = 1.092 (*SE* = 0.493), *t*(841) = 2.214, *p* < 0.028, 95% CI [0.124, 2.060], Model Δ*R*^2^ (due to the three-way interaction) < 0.004, overall Model *R*^2^ = 0.344, *p* < 0.001. The Johnson–Neyman values showed a negative relation between FAB and anxiety that began as significant, decreased as IAPSA increased and was last significant at IAPSA ratings of 2.803 (before the first quintile), and it inverted and became significantly positive at IAPSA ratings of 4.171 (beyond the fifth quintile) and continued to increase from that point (i.e., Figure 11a–e).

When examining the three-way interaction of initial event affect, IAPSA (problem-solving attitudes), and stress on fading affect while controlling for initial affect intensity and participant, the Hayes Process Model 3 [61] showed that all the main effects and two-way interaction between IAPSA, stress, and initial event affect were significant; B = 1.615 (*SE* = 0.399), *t*(841) = 4.049, *p* < 0.001, 95% CI [0.832, 2.398], Model Δ*R*^2^ (due to the three-way interaction) < 0.013, overall Model *R*^2^ = 0.353, *p* < 0.001. The Johnson–Neyman values displayed a negative relation between FAB and stress that began statistically significant and decreased as IAPSA increased until it was last significant at IAPSA ratings of 3.000, and then that negative relation inverted and became significantly positive at IAPSA ratings of 3.528 and continued to increase from that point (i.e., Figure 11a–e).

The Hayes Process Model 3 (Hayes, 2022 [61]) examined the three-way interaction of initial event affect, positive problem-solving attitudes (IAPSA), and poor sleep on fading affect while controlling for initial affect intensity and participant. The model showed that all main effects and two-way interactions were significant, except for the main effect of IAPSA and the initial event affect and IAPSA interaction. Moreover, the model demonstrated a significant three-way interaction between IAPSA, poor sleep, and initial event affect; B = 1.058 (*SE* = 0.528), *t*(841) = 2.002, *p* < 0.046, 95% CI [0.076, 2.284], Model Δ*R*^2^ (due to the three-way interaction) = 0.003, overall Model *R*^2^ < 0.347, *p* < 0.001. The Johnson–Neyman values demonstrated a negative relation between FAB and poor sleep that began as significant and stopped being significant at IAPSA ratings of 3.283 (at the precipice of the third quintile), and the relation flipped and increased with IAPSA ratings from that point without becoming significant (i.e., Figure 11a–e).

##### Only Thinking Rehearsals Mediated Two Three-Way Interactions

Next, we examined the conditional indirect effects of initial event affect on fading affect for positive problem-solving attitudes (IAPSA) and neuroticism, depression, stress, anxiety, and poor sleep through rehearsal ratings using the Process Model 11 (Hayes, 2022 [61]). The relation between the three-way interaction involving initial event affect, IAPSA, and depression on fading affect was intervened by thinking rehearsal ratings at the first and second quintiles of depression across the first and second quintiles of IAPSA scores, and the first and fifth quintiles of depression at the third quintile of IAPSA. The three-way interaction of initial event affect, IAPSA, and stress on fading affect was intervened by thinking rehearsal ratings at the first quintile of stress across the first, second, and third quintiles of IAPSA, and the fifth quintile of stress at the second quintile of IAPSA. No other mediation effects were demonstrated.

#### 2.2.3. Discussion

The robust FAB effect shown in Study 2 was nearly the same size as the FAB effect in Study 1 with half the event sample. Therefore, Study 2 replicated and extended the Study 1 problem-solving FAB effect from an in-person context before and after the COVID-19 pandemic to an online context during the pandemic, showing internal and external validity. However, Study 2 did not replicate the effect in Study 1 that problem-solving events produced larger FAB than non-problem-solving events. Whereas FAB was larger for problem-solving events than non-problem-solving events when no controls were enlisted, no event difference in FAB was found when initial affect intensity was statistically controlled to eliminate regression-to-the-mean as an explanation. Much like Study 1, Study 2 showed that talking rehearsals and the combined thinking and talking rehearsals positively predicted FAB, which supported past research showing that social sharing positively predicted FAB (i.e., Ritchie et al., 2006 [34]; Skowronski et al., 2004 [35]; Walker et al., 2009 [36]).

Interestingly, neuroticism positively predicted FAB, which was unexpected because unhealthy variables typically negatively predict FAB. Nevertheless, two precedents exist for this result: Gibbons and Bouldin (2019) [31] found that depression positively predicted FAB at high levels of Internet addiction, and Gibbons et al. (2023) [28] discovered that neuroticism, hypochondria, and average daily thinking and talking about COVID-19 positively predicted FAB at high levels of coronaphobia. Similarly, Study 2 revealed that neuroticism, depression, anxiety, stress, and poor sleep positively predicted FAB at medium to high levels of positive problem-solving attitudes. Whereas the unexpected positive relations between unhealthy variables and FAB in other studies depended on medium to high levels of a moderator, the positive relation between neuroticism and FAB in Study 2 did not. These findings could mean that Study 2 participants experienced higher than normal levels of positive problem-solving attitudes, possibly because the experiment advertised problem-solving in its title and attracted individuals drawn to problem-solving. This speculation was supported by first-quintile IAPSA ratings of 3.067 (1 to 5 scale). Alternatively, Study 2 participants could have experienced high levels of some other unmeasured variable that combined with neuroticism above floor levels and led to high FAB. Although the first possibility could be easily tested, testing the second possibility would demand divergent creativity.

Relatedly, neuroticism, depression, anxiety, stress, and poor sleep exhibited negative or no relations to FAB at low levels of positive problem-solving attitudes, and positive relations to FAB at medium to high levels of positive problem-solving attitudes. These findings suggest that positive problem-solving attitudes are powerful enough to reduce and reverse the harmful effects of unhealthy variables on emotion regulation in the form of FAB. Therefore, future research should continue to investigate the ability of this problem-solving variable to promote emotional health. The complex moderation effects were mediated by thinking rehearsals. Specifically, thinking rehearsals partially mediated the interactive effects of initial event affect, depression, and positive problem-solving attitudes on fading affect at 6 quintiles of the 25 possible quintiles of the interaction. Similarly, thinking rehearsals partially mediated the interactive effects of initial event affect, stress, and positive problem-solving attitudes on fading affect at 4 quintiles of the 25 possible quintiles of the interaction. No other mediation effects were found.

As depression and stress produced the three-way interactions with the largest effect sizes, this factor might have helped to account for the mediation effects observed in Study 2. In addition, the online format and the timing of the study could have accounted for the finding that thinking rehearsals were the only rehearsal type to mediate any three-way interactions in Study 2. The online format could have attracted individuals who enjoyed interacting on the Internet, and these individuals could have preferred to mentally rehearse their events, or they simply engaged in more mental rehearsals than social rehearsals. Similarly, the study was conducted in the middle of a pandemic, and, hence, the participants in Study 2 may have engaged in more mental rehearsals than social rehearsals due to the isolation produced by the pandemic.

## 3. General Discussion

Although Study 1 and Study 2 used the same general population pool, they also used different data collection methods (in-person and online) as well as different data collection times (before and after the COVID-19 pandemic for Study 1 and during the COVID-19 pandemic for Study 2). The common findings across the two studies become very important considering the strong differences between the two studies. Both studies showed robust FAB effects, and they both demonstrated positive relations between healthy variables and FAB as well as negative relations between unhealthy variables and FAB. These results support theoretical arguments that biological, cognitive, and emotional resources are mobilized in response to and reduce the harmful effects of unpleasant autobiographical events [22] and encourage avoidance of these events in the future, enhancing perceptions of the self [51,52]. In addition, rehearsals involving talking positively predicted the FAB in both studies, which supports the findings of past research [34,35,36]. These results highlight the importance of talking rehearsals for predicting FAB effects. Based on these findings, future FAB research in our laboratory will prioritize, measure, and evaluate the impact of talking rehearsals potentially at the exclusion of the other rehearsal measures used in the current study.

As mentioned in the discussion of Study 2, unhealthy variables did not predict or they negatively predicted FAB at low levels of a moderating variable in Study 2, positive problem-solving attitudes, a different moderating variable in the Gibbons and Bouldin (2019) [31] study on video games, Internet gaming addiction, and another moderating variable in the Gibbons et al. (2023) study on COVID-19, coronaphobia. Although these three different moderators produced similar outcomes by reducing and inverting negative relations of unhealthy variables and FAB, positive problem-solving attitudes are healthy/helpful/pleasant, and both Internet gaming addiction and coronaphobia are unhealthy/harmful/unpleasant. Therefore, a common explanation is unlikely to account for the reason why these three variables moderated the relations of predictor variables and FAB.

One simple explanation for the moderating effects of the positive problem-solving attitudes variable is that it is powerful enough to transform harmful emotional effects into healthful ones. Conversely, the moderating effects of Internet gaming addiction and coronaphobia are not as easily explained. For example, Gibbons and Bouldin (2019) [31] suggested that individuals with high Internet gaming addiction (i.e., gamers) accepted depression as an expected outcome from challenging games, which they pursued, consequently leading to high FAB. Alternative explanations could certainly exist. In addition, Gibbons et al. (2023) [28] suggested that coronaphobia could have helped participants habituate to aversive emotional conditions and learn to expect and appreciate them, which enhanced emotional regulation. Alternatively, coronaphobia could have distracted and confused participants to such a strong degree that they processed unhealthy variables and emotions in an emotionally healthy way.

Although the common results for Study 1 and Study 2 are important for internal validity and external validity in the form of generalization, the different results across the two studies are informative as well. For example, the FAB was larger for problem-solving events than for non-problem-solving events in Study 1, but this effect was not replicated in Study 2. Sample size might have accounted for this difference if the other effects in Study 2 were not so numerous, and the online format in Study 2 was unlikely to have been responsible for the absence of the event type differences due to the same overall population. Therefore, the difference in the timing of the studies relative to the COVID-19 pandemic might account for those differences. Specifically, pandemic-related issues were likely the primary problems to solve during the pandemic, overshadowing other problems at that time for everyone. In contrast to problem-solving events, positive problem-solving attitudes seem to have been more important in reducing and inverting the harmful emotional effects of unhealthy variables during the COVID-19 pandemic, based on the many three-way interactions in Study 2, than before or after that time period. Positive problem-solving attitudes seem to have provided a particularly protective shield for participants during the pandemic describing their events, thoughts, and emotions online. Future research should examine whether interventions can increase positive problem-solving attitudes along with emotional regulation in the form of FAB.

The fact that positive PANAS and emotional intelligence interacted to predict FAB in Study 1 but not Study 2 may indicate that positive feelings in the moment may not have been as abundant and/or as important during the COVID-19 pandemic as they were before and after that time period or they could be explained by a lack of power in Study 2. The literature on emotions and the COVID-19 pandemic supports the contention that people experienced many unpleasant and unhealthy emotions and fewer pleasant and healthy emotions during that time period (e.g., [28]). However, the multitude of significant, complex effects in Study 2 suggest that low power was an unlikely explanation for the absence of an interactive effect, involving initial affect, positive PANAS, and emotional intelligence, on fading affect.

As stated in the discussion of Study 2, rehearsals involving talking were successful in positively predicting FAB in both studies. However, Study 1 showed that all rehearsal ratings partially mediated the two significant three-way interactions and that thinking rehearsals were the only rehearsal ratings to mediate, albeit partially, three-way interactions in Study 2. Based on the notion that mediation effects, full or partial, for three-way interactions are more important than two-way interactions showing rehearsals predicting FAB, future research in our laboratory will continue to consider thinking rehearsals as a viable variable to account for complex, three-way FAB interaction effects. As past research in our lab asked participants to provide the frequency that they thought or talked about an event and those rehearsal ratings successfully mediated complex, three-way FAB interaction effects, future research should consider using them again. One option is to ask participants to provide thinking or talking rehearsals before talking rehearsals.

Like any study, the current study unfortunately included limitations. Specifically, we used a broad definition for problem-solving events. Participants were told that such an event involved a solution, which could have included interpersonal, complex, work-related, mechanical, electrical, financial, or sports-related events. Future research could focus on one of these areas, but students experience academic problems more than any other area. Interpersonal problems should be another frequently experienced area for students. Of course, online experiments using MTurk to collect data from individuals outside of college campuses could focus on any of the previously mentioned areas.

In summary, robust FAB effects were demonstrated in the context of problem-solving, which yielded positive relations between the FAB and healthy variables as well as negative relations between FAB and unhealthy variables. These latter results support the mobilization-minimization hypothesis and self-enhancement theories. In addition, high levels of positive problem-solving attitudes seemingly reduced and inverted negative relations of unhealthy variables to FAB, which demonstrated emotional regulation. Furthermore, rehearsals mediated the complex three-way interactions involving positive problem-solving attitudes, unhealthy variables, and the FAB, which is a relative success considering that we have not shown these effects in our lab for several years after following reviewers’ suggestions. In conclusion, the FAB extends well to the context of problem-solving, but the critical lesson in this paper may be that the gift of positivity is bestowed upon people who perceive every problem as a solution waiting to happen.

## Figures and Tables

**Figure 1 behavsci-14-00806-f001:**
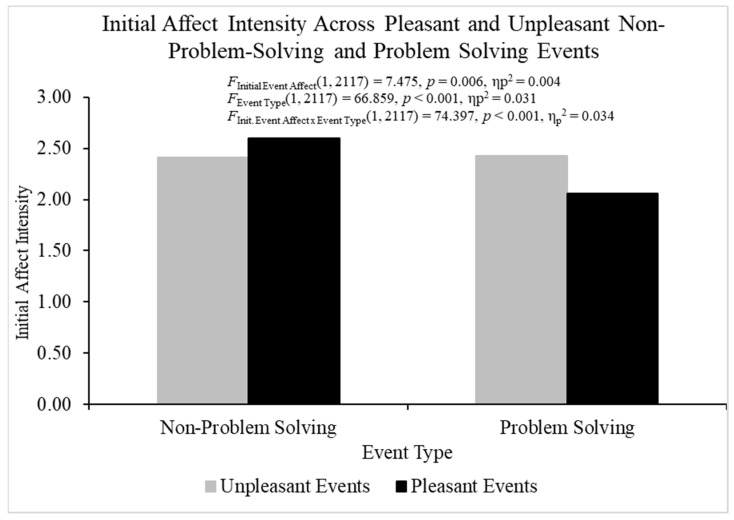
Initial affect intensity of pleasant and unpleasant non-problem-solving and problem-solving events in Study 1.

**Figure 2 behavsci-14-00806-f002:**
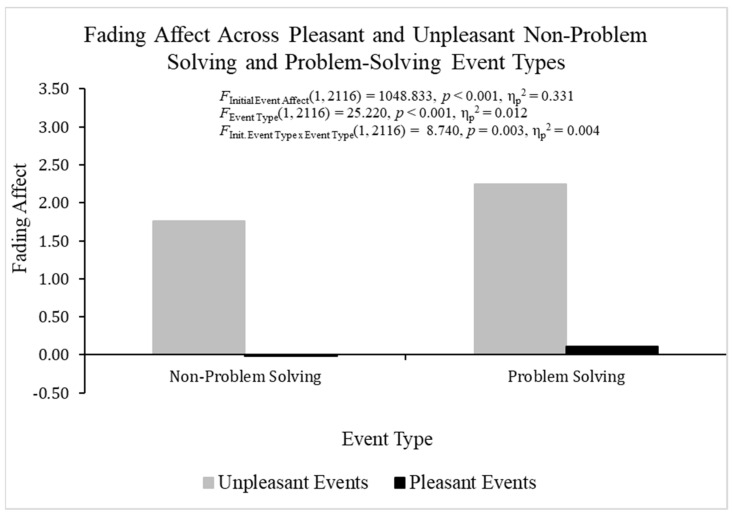
Fading affect of pleasant and unpleasant non-problem solving and problem-solving events in Study 1.

**Figure 3 behavsci-14-00806-f003:**
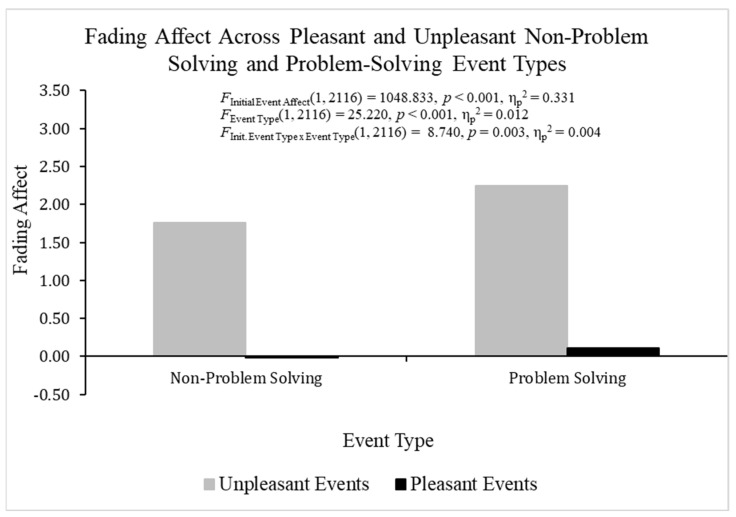
Fading affect of pleasant and unpleasant events across quintiles of talking rehearsals in Study 1.

**Figure 4 behavsci-14-00806-f004:**
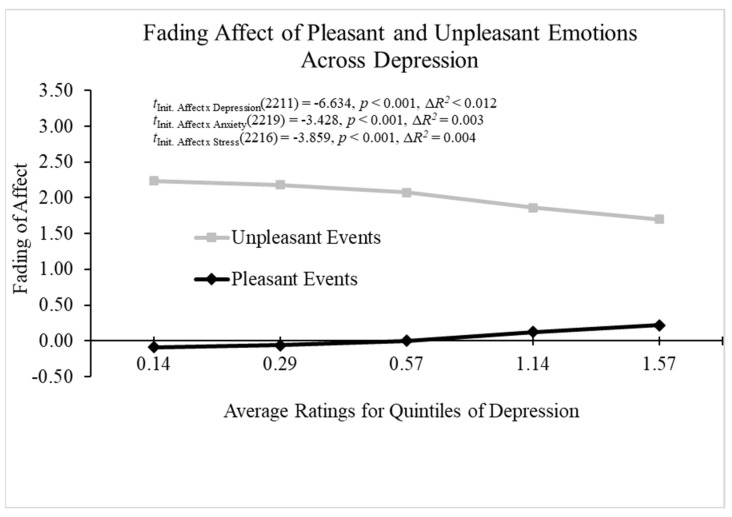
Fading affect of pleasant and unpleasant events across quintiles of depression in Study 1.

**Figure 5 behavsci-14-00806-f005:**
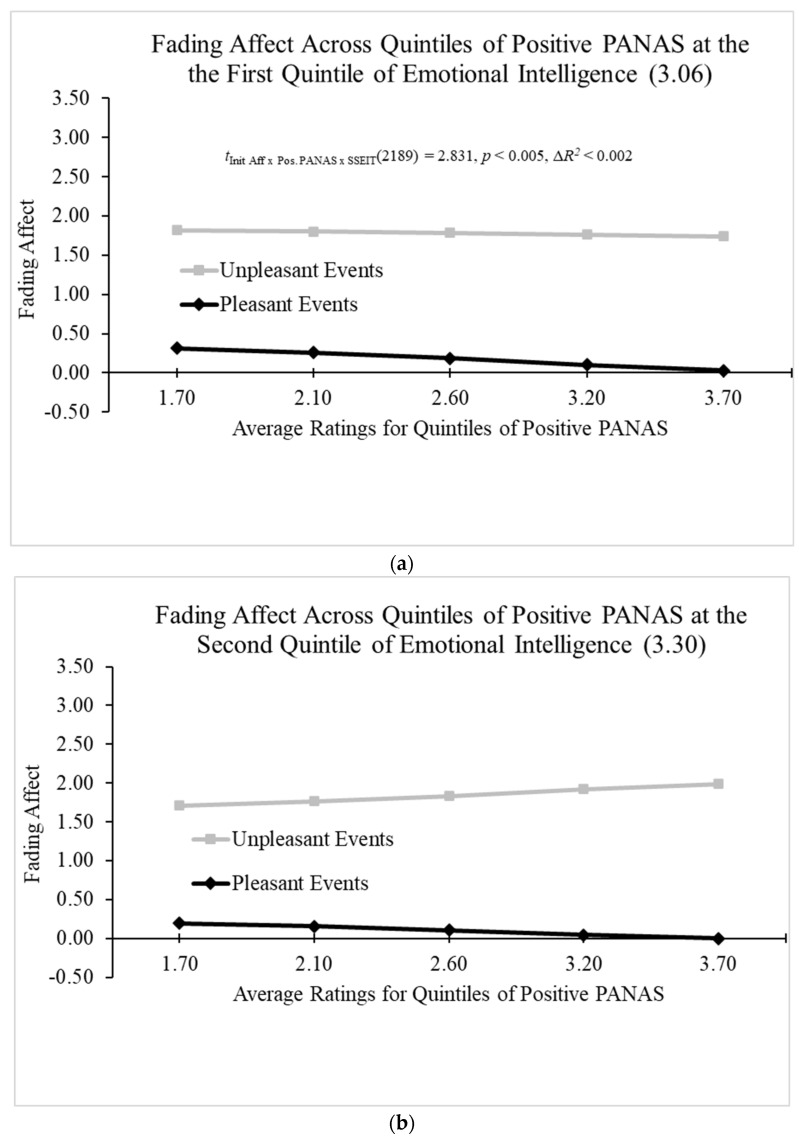
(**a**): Fading affect of pleasant and unpleasant events across quintiles of positive PANAS at the first quintile (3.061) of emotional intelligence (SSEIT) in Study 1. (**b**): Fading affect of pleasant and unpleasant events across quintiles of positive PANAS at the second quintile (3.303) of emotional intelligence (SSEIT) in Study 1. (**c**): Fading affect of pleasant and unpleasant events across quintiles of positive PANAS at the third quintile (3.546) of emotional intelligence (SSEIT) in Study 1. (**d**): Fading affect of pleasant and unpleasant events across quintiles of positive PANAS at the fourth quintile (3.788) of emotional intelligence (SSEIT) in Study 1. (**e**): Fading affect of pleasant and unpleasant events across quintiles of positive PANAS at the fifth quintile (4.000) of emotional intelligence (SSEIT) in Study 1.

**Figure 6 behavsci-14-00806-f006:**
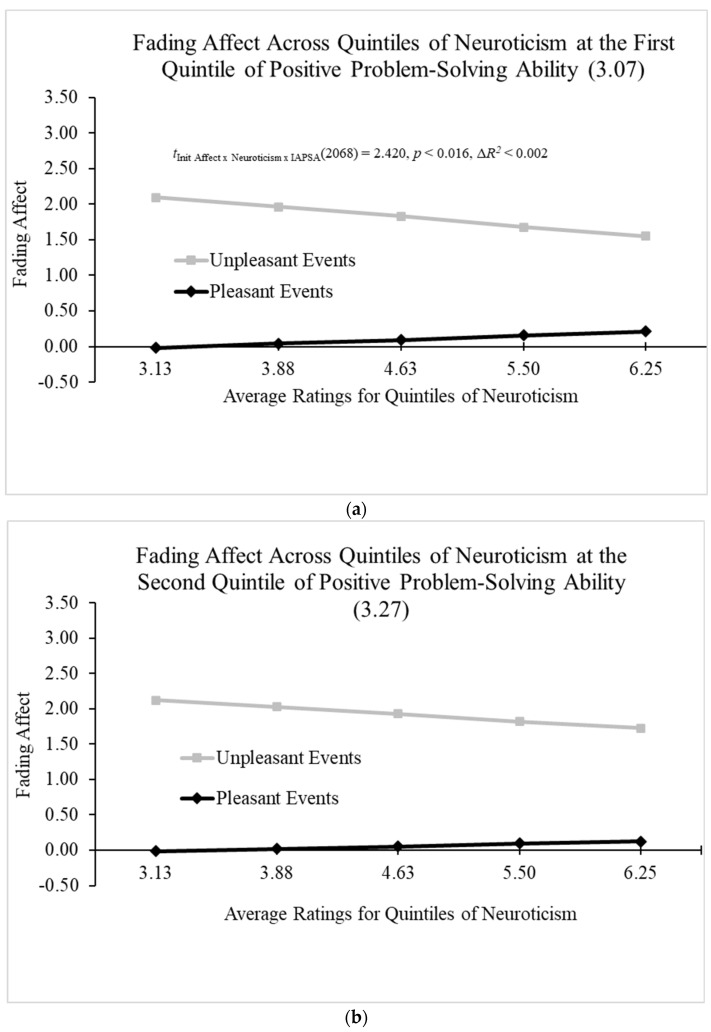
(**a**): Fading affect of pleasant and unpleasant events across quintiles of neuroticism at the first quintile (3.067) of positive problem-solving attitudes (IAPSA) in Study 1. (**b**): Fading affect of pleasant and unpleasant events across quintiles of neuroticism at the second quintile (3.267) of positive problem-solving attitudes (IAPSA) in Study 1. (**c**): fading affect of pleasant and unpleasant events across quintiles of neuroticism at the third quintile (3.433) of positive problem-solving attitudes (IAPSA) in Study 1. (**d**): fading affect of pleasant and unpleasant events across quintiles of neuroticism at the fourth quintile (3.633) of positive problem-solving attitudes (IAPSA) in Study 1. (**e**): fading affect of pleasant and unpleasant events across quintiles of neuroticism at the fifth quintile (3.833) of positive problem-solving attitudes (IAPSA) in Study 1.

**Figure 7 behavsci-14-00806-f007:**
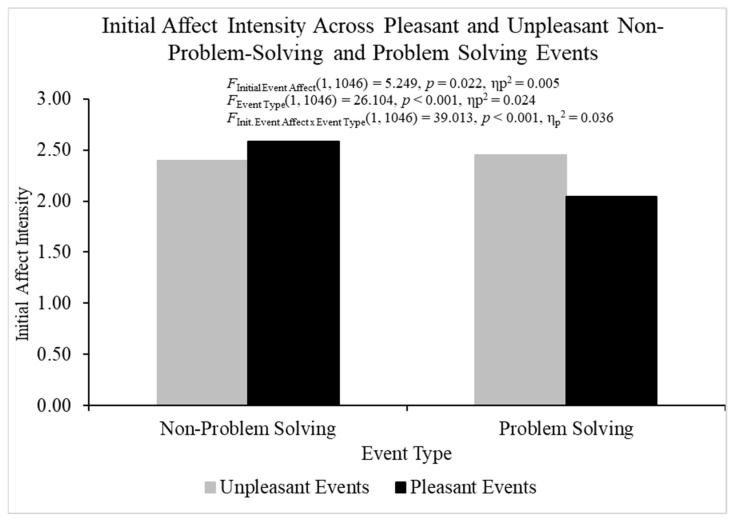
Initial affect intensity of pleasant and unpleasant non-problem-solving and problem-solving events in Study 2.

**Figure 8 behavsci-14-00806-f008:**
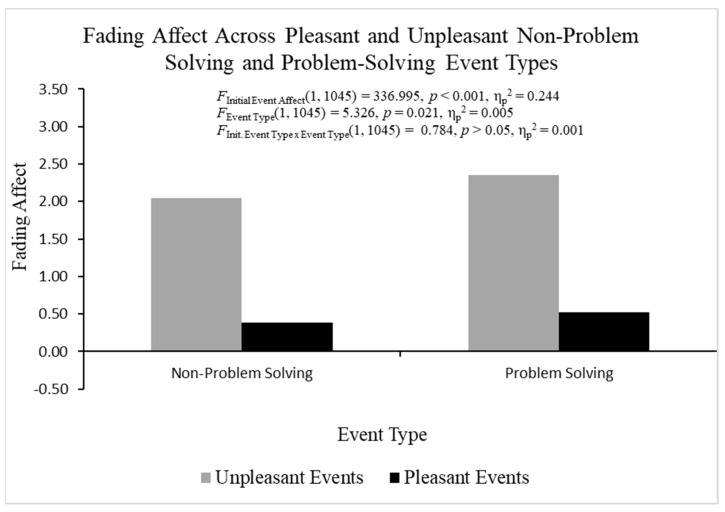
Fading affect of pleasant and unpleasant non-problem-solving and problem-solving events in Study 2.

**Figure 9 behavsci-14-00806-f009:**
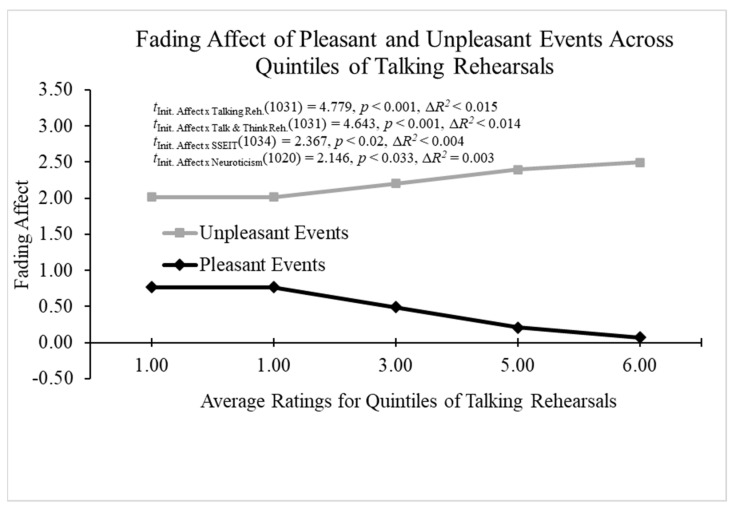
Fading affect of pleasant and unpleasant events across quintiles of talking rehearsals in Study 2.

**Figure 10 behavsci-14-00806-f010:**
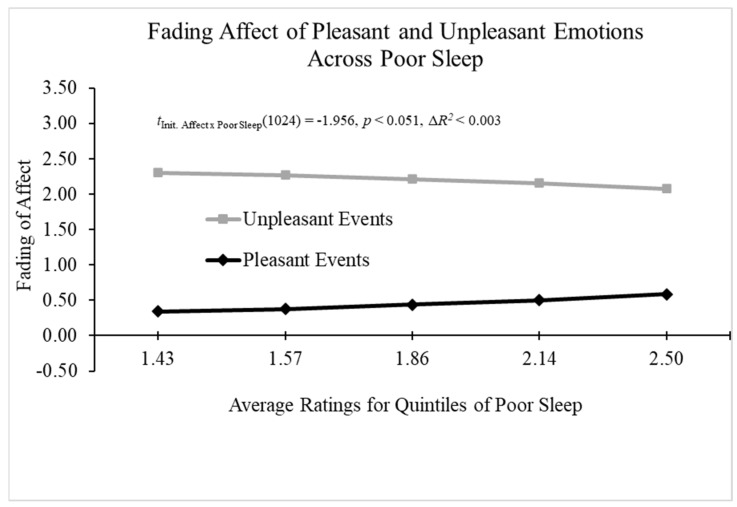
Fading affect of pleasant and unpleasant events across quintiles of poor sleep in Study 2.

**Figure 11 behavsci-14-00806-f011:**
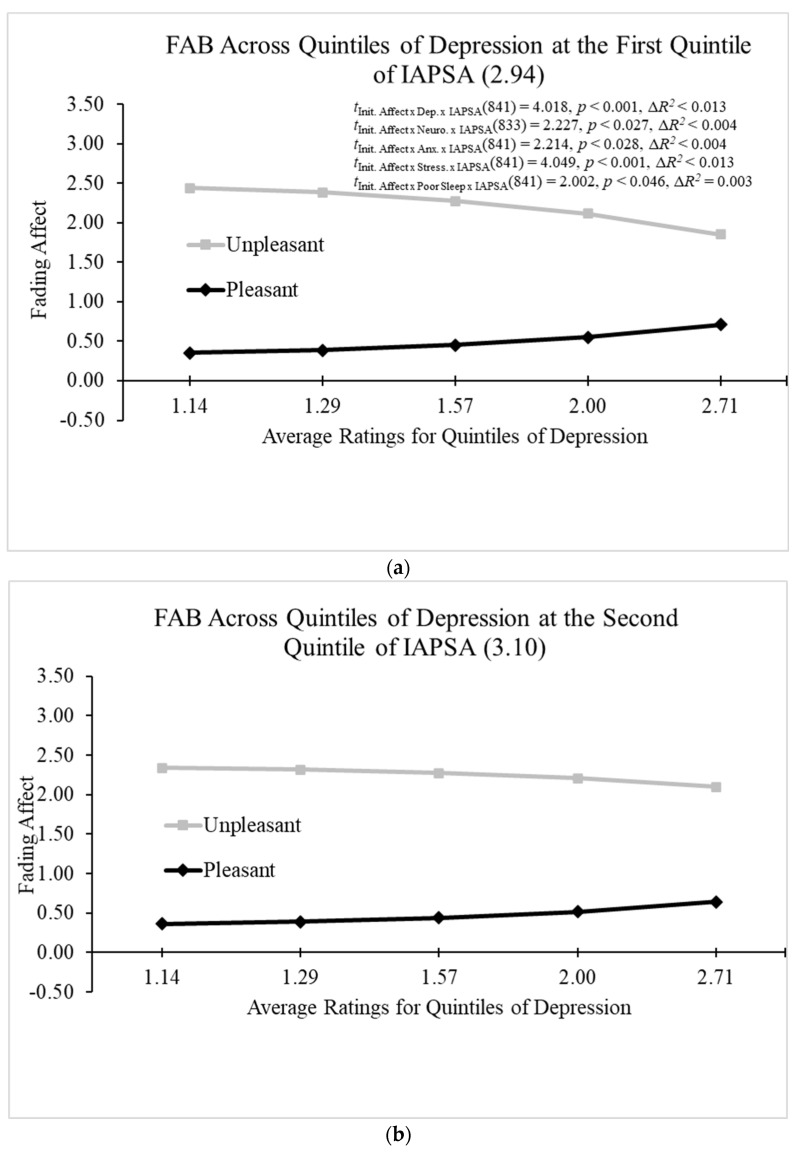
(**a**): Fading affect of pleasant and unpleasant events across quintiles of depression at the first quintile (2.936) of positive problem-solving attitudes (IAPSA) in Study 2. (**b**): Fading affect of pleasant and unpleasant events across quintiles of depression at the second quintile (3.097) of positive problem-solving attitudes (IAPSA) in Study 2. (**c**): Fading affect of pleasant and unpleasant events across quintiles of depression at the third quintile (3.290) of positive problem-solving attitudes (IAPSA) in Study 2. (**d**): Fading affect of pleasant and unpleasant events across quintiles of depression at the fourth quintile (3.548) of positive problem-solving attitudes (IAPSA) in Study 2. (**e**): Fading affect of pleasant and unpleasant events across quintiles of depression at the fifth quintile (3.839) of positive problem-solving attitudes (IAPSA) in Study 2.

## Data Availability

Data will be provided upon reasonable request.

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
