# Peer review of "In-Person and Online Studies Examining the Influence of Problem Solving on the Fading Affect Bias"

_behavsci, 2024, doi:10.3390/bs14090806_

Round 1

Reviewer 1 Report

Comments and Suggestions for Authors

I think this paper makes a strong contribution to the literature: The studies are generally well-motivated in the Introduction, it is statistically and methodologically sound, and the authors draw reasonable and well-founded conclusions from their findings. I therefore only have one small comment, namely, that I think the rationale for extending the in-person results in Study 1 to the online method of Study 2 is somewhat undermotivated in the paper; it's not clear to me why the authors thought it was important to look at the issue through both methods, or why they thought their findings might vary across the in-person method versus the online method. Therefore, I think the paper would be stronger with a more developed explanation of the rationale here. 

Author Response

Reviewer 1 said “I think the rationale for extending the in-person results in Study 1 to the online method of Study 2 is somewhat undermotivated in the paper; it's not clear to me why the authors thought it was important to look at the issue through both methods, or why they thought their findings might vary across the in-person method versus the online method. Therefore, I think the paper would be stronger with a more developed explanation of the rationale here.”  We agree with Reviewer 1 that we did not justify the specific type of replication in Study 2.  Therefore, we provided that justification in the introduction of Study 2 before the hypotheses.  We thank Reviewer 1 for making this comment because the justification of Study 2 was necessary, and it may not have been provided if not for Reviewer 1.   

Reviewer 2 Report

Comments and Suggestions for Authors

Review of In-Person and Online Studies Examining the Influence of Problem-Solving on the Fading Affect Bias

The authors reported greater fading for problem solving events in study 1 but not in study 2 for college students in this manuscript. Authors say sample size and timing of the studies (various time points during the COVID 19 pandemic, etc.) might have contributed to disparate results. Grit and other “healthy” variables related to greater FAB and depression and other psychological distress variables were inversely related to FAB. Talking and thinking rehearsals mediated these relationships with FAB.

Authors included an adequate sample size for in-person participants (and sample size for online participants was smaller than in study 1) in the two studies.  

Line 462 correct ANCOVA.

Line 589 authors reference Figure 5, but there are several figures before the reference and they need to be delineated properly.

Line 616 same comment as above.

Line 833—figures need to be individually and consecutively numbered.

Line 950 “more mentally” should be edited to make better sense to the reader.

The manuscript contributes to our understanding of the relationships between problem solving and non-problem solving events and FAB and after modifications as described above, I recommend publishing this important contribution to the literature.

Author Response

Reviewer 2 made several comments.  Specifically, Reviewer 2 said that “the authors included an adequate sample size for in-person participants (and sample size for online participants was smaller than in study 1) in the two studies.”  We agree.  Reviewer 2 said “Line 462 correct ANCOVA.”  We are very happy that Reviewer 2 made this comment.  The initial affect intensity analyses in both studies suggested that we needed to control for initial affect intensity when analyzing the fading affect.  Therefore, we changed the analytic strategy in Study 1, and we changed the results in both studies to address this issue.  The analytic strategy and results are aligned, and they make much more sense now.  Reviewer 2 said “Line 589 authors reference Figure 5, but there are several figures before the reference and they need to be delineated properly. Line 616 same comment as above.”  At the end of the paper, all the Figure Captions were placed in order and all the Figures were placed in order after the Figure Captions.  The editor placed all 5 Figure Captions (Figures 5a-5e) at the bottom of Figure 5e to represent one figure, which is understandable, but confusing.  Instead, we created a figure caption for each graph/picture of Figure 5.  We think each figure caption should be placed under each graph/picture.  If the editor wants to place an additional figure caption after the figure caption for Figure 5e to describe all graphs/pictures in Figure 5, it should say “Fading Affect of Pleasant and Unpleasant Events Across Quintiles of Positive PANAS at Each Quintile of Emotional Intelligence (SSEIT) in Study 1”.  If the editor cannot or does not want to make these changes, we can make that change before the paper is sent back to reviewers or the copy editor assuming acceptance.  We simply need to know before the paper is sent to the reviewers or copy editor if they cannot or will not make the change.  Reviewer 2 said “Line 833—figures need to be individually and consecutively numbered.”  To address this issue, we need to know the editor’s plans.  We need to know that the editor will make the necessary changes to the figure captions and figures or direct us to change them.  We also need to know if the editor wants us to put figure numbers in each figure.  The editor can put these numbers in the figures or we can do it but we need to know if and how the editor wants this change made.  For example, we could put the Figure number (e.g., Figure 1) in the title with a colon following it and the title following the colon.  Reviewer 2 said “Line 950 “more mentally” should be edited to make better sense to the reader.”  We changed the wording to enhance the legibility of this sentence.